# Prevalence and determinants of depression, anxiety, and perceived stress among patients with chronic comorbidity attending outpatient clinics in Addis Ababa during the COVID-19 pandemic

Etsegenet Dege Dires[1], Ayto Addisu Negash[2*], Getachew W/Yohannes[3], Firdos Ali Mohammed[4], Muna Ahmed Redi[5], Birhanu Fenta Tsega[6], Agazhe Melaku Mihiretie[7], Sifessa Dessalegn Demshasha[8], Tekiy Markos Bedore[2], Yitayew Ewnetu Mohammed[9]

1 Department of Family Medicine, St. Paul's Hospital Millennium Medical College, Addis Ababa, Ethiopia, 2 Department of Emergency Medicine and Critical Care, Werabe Comprehensive Specialized Hospital, Werabe University, Werabe, Ethiopia, 3 Department of Public Health Head, Yekatit 12 Hospital Medical College, Addis Ababa, Ethiopia, 4 Department of Internal Medicine, Zewditu Memorial Hospital, Addis Ababa, Ethiopia, 5 Department of Family Medicine Tirunesh Beijing General Hospital, Addis Ababa, Ethiopia, 6 Department of Family Health Team and Health Extension of Kality Health Center, Addis Ababa, Ethiopia, 7 Department of Clinical Radiology, St Paul's Hospital Millennium Medical College, Addis Ababa, Ethiopia, 8 Department of Internal Medicine, St. Paul's Hospital Millennium Medical College, Addis Ababa, Ethiopia, 9 Department of internal Medicine, Werabe Comprehensive Specialized Hospital, Werabe University, Werabe, Ethiopia

* aytoaddisu19@gmail.com

## Abstract

The COVID-19 pandemic has disproportionately affected individuals with chronic conditions, increasing risks of severe illness and psychological distress. In Ethiopia, limited healthcare access and disrupted social support have heightened depression, anxiety, and stress, yet evidence on this population remains scarce. This study assessed the prevalence and determinants of depression, anxiety, and perceived stress among patients with chronic comorbidity in Addis Ababa during the COVID-19 pandemic. A cross-sectional study was conducted from April to June 2021 at St. Paul's Hospital Millennium Medical College and Black Lion Hospital. A total of 437 adults were selected using systematic random sampling, and 426 participants were included in the analysis. Data were collected using Hospital Anxiety and Depression Scale (HADS), the 10-item Perceived Stress Scale (PSS-10-C), and Oslo-3 Social Support Scale. Logistic regression analysis was used to identify factors associated with psychological distress. The mean age of participants was 49.6 years; 52.1% were female. Hypertension, heart disease, and diabetes were the most common chronic conditions, and 46.5% of participants had multiple comorbidities. Severe perceived stress was reported by 45.5% of participants, while depression and anxiety were identified in 39.7% and 26.5% of participants, respectively, based on combined

**Data availability statement:** You can access the updated dataset and supplemental materials through the following link: https://doi.org/10.6084/m9.figshare.31143931.

**Funding:** The authors received no specific funding for this work.

**Competing interests:** The authors have declared that no competing interests exist.

proportion of participants with borderline and abnormal levels on the HADS. Higher psychological distress was associated with younger age, multiple comorbidities, female sex, unmarried status, low education, and poor social support. The findings indicate that the COVID-19 pandemic has amplified mental health challenges among patients with chronic illnesses in Addis Ababa. Integrating targeted psycho-social interventions and conducting longitudinal studies are essential to mitigate long-term mental health impacts of the pandemic.

## Introduction

### Background of study

The corona-virus disease (COVID-19) emerged in December 2019 and rapidly became a global pandemic, affecting millions worldwide. By December 2020, over 61.8 million confirmed cases and 1.4 million deaths had been reported globally, including 110,554 cases and 1,709 deaths in Ethiopia, one of the low-income countries heavily impacted by the pandemic [1,2]. COVID-19 has had a disproportionate impact on individuals with chronic medical illnesses, who are more vulnerable to severe infection, frequent hospitalization, and death [3–6].

Beyond its physical effects, the pandemic has created major psychological stressors, particularly for those with chronic comorbidity. Social distancing and lockdown measures have disrupted social support systems and access to routine medical care, exacerbating existing mental health challenges [7–9]. Isolation and the loss of physical contact, once a vital form of emotional connection, have intensified feelings of distress [8], Limited access to online or e-mental-health services in countries like Ethiopia further restricts coping options, heightening the psychological toll [9].

Even before the pandemic, mental health disorders were common among individuals with chronic illnesses. Depression affected 4–5% of the general population [10] with higher prevalence among those with medical, social, or psycho social risk factors [11]. Studies report that 27–40% of patients with chronic obstructive pulmonary disease (COPD) experience clinically relevant depression [12,13] and up to 36% present with significant anxiety symptoms [13]. Among people with diabetes mellitus (DM), the prevalence of mental disorders ranges from 20% to 55% [9], while 13–17% of cancer patients exhibit depressive symptoms. Such co-morbid psychological conditions are linked to poorer outcomes, including increased morbidity, suicide risk, and reduced quality of life.

During the pandemic, these psychological burdens among patients with chronic diseases have worsened due to fear of infection, uncertainty, self-isolation, and disruptions in healthcare access. Additional stressors, such as financial hardship, unemployment, and overwhelming news exposure, have further contributed to anxiety and depression [14]. Untreated psychological distress, anxiety, and depression may lead to non-adherence to life style modifications and medical treatment, reduced survival, higher healthcare costs, and decreased quality of life [15]. Moreover COVID-19 related stress has even been associated with increased risk of stress cardiomyopathy

[16], and worsened pulmonary morbidity [17]. Anxiety and depression also hinder lifestyle modification [18], contribute to hypertension [19], and negatively affect survival in cancer patients [20]. Ensuring continuity of care for patients with chronic diseases is thus vital to prevent an escalation of non-COVID-19-related morbidity and mortality [21].

Ethiopia, with over 116 million people [22], faces major healthcare challenges, including a low doctor-patient ratio [23,24], high treatment costs, and widespread health illiteracy. Social networks, crucial for coping, were disrupted by pandemic restrictions, likely worsening mental health among low-income patients with chronic illnesses.

Despite these compounded challenges, there remains limited research on the psychological impact of the COVID-19 pandemic on this population, making an understanding of the prevalence and determinants of depression, anxiety, and perceived stress essential to inform evidence-based interventions and integrate mental health support into routine chronic disease management.

## Objective of the study

This study aims to assess the prevalence and determinants of depression, anxiety, and perceived stress among patients with chronic comorbidities attending outpatient clinics in Addis Ababa during the COVID-19 pandemic.

## Materials and methods

### Study area, duration, and design

A facility-based cross-sectional study was conducted at Black Lion Hospital and St. Paul's Hospital Millennium Medical College, Addis Ababa, Ethiopia. Black Lion Hospital and St. Paul's Hospital Millennium Medical College (SPHMM) are two of the largest hospitals providing specialty care to patients from Addis Ababa and nationwide [25,26]. The data collection period was from April 10, 2021 to June 30, 2021GC, which corresponds to the early post–first-wave phase of the COVID-19 pandemic in Ethiopia. During this time, national cases peaked in April and declined thereafter, and vaccination had only recently begun, with most of the population still unvaccinated. This context is important for interpreting the findings.

### Population

All patients with chronic medical patients aged 18 and above who were on follow up treatment at St. Paul's hospital millennium medical college and black lion hospital outpatient clinics (renal dialysis, oncology, pulmonology, endocrinology, and cardiology clinic) were the source population. All patients with chronic medical conditions who were on follow up treatment at SPHMM and BLH outpatient departments during the study period were the study population. Patients who declined to be interviewed, and those with existing mental health problems already on treatment at psychiatry OPD were excluded from the study.

**Eligibility criteria.** Inclusion Criteria: All patients with comorbidity aged 18 and above who were on follow up treatment at St. Paul's hospital millennium medical college and black lion hospital OPD during the study period were included.

Exclusion Criteria: Patients who declined to be interviewed were excluded. In addition, critically ill patients or those experiencing acute exacerbation's of their chronic conditions (such as DKA, HHS, acute asthma attacks, or COPD exacerbation's) were excluded because acute medical events themselves are significant psychological stressors, making it difficult to accurately attribute psychological distress specifically to the COVID-19 pandemic. Patients with preexisting mental health conditions who were already receiving follow-up care at the psychiatry outpatient department were also excluded to avoid confounding and to ensure that the psychological outcomes assessed reflected new-onset or pandemic-related effects. While these exclusions were necessary for methodological clarity, we acknowledge that they may introduce selection bias and potentially underestimate the true burden of psychological distress among the broader chronic-illness population.

**Study variables.** Dependent variables: Depression, Anxiety and Perceived stress level.

Independent variables:

Socio-demographic variables**:** Age, sex, marital status, place of residence, patient's occupation, educational level, family size, number of house room, being primary care giver for children or other relatives.

Clinical characteristics and risk assessment of the study participants**: -** type of chronic diseases, age of onset of chronic illness, duration of chronic disease, number of co-morbidities, physical(social) activity, substance use, presence of respiratory symptoms in the past two weeks, use of a face mask and sanitizer, contact history with known covid-19 case and having social support.

Environmental related concern: - the wide community spread of covid-19**,** inadequate national response**,** working in a place where one is susceptible to contract COVID-19 and restriction to engage in several religious festivals, get-together.

Socioeconomic related concern: - Self-isolation or quarantine because of COVID -19 infection, concern about infecting others, difficulty in obtaining help, social support or availability of appropriate care needed, difficulty obtaining food, health care, medicine, or essentials and loss of job or loss of income related to COVID-19.

Health related concern: - Postponement or cancellation of treatment, changes in their health care trajectory, the Possibility that their co-morbid illness may progress or be less likely to be cured due to changes in treatment, getting sick from exposure to COVID-19 and subsequent poorer outcomes during the COVID-19 pandemic and concerned about difficulties in managing their co-morbid illness if they contract COVID-19.

## Sampling

Sampling size determination: A single population proportion formula [n = (Z a/2)$^2$ P(1-P)/d$^2$] was used to determine the required sample size. The assumptions used to calculate the sample size were: 95% level of confidence interval (Z $\alpha/2 = 1.96$), 5% margin of error. Published data in Ethiopia depicted the prevalence of abnormal psychological impact was 22.8% [27], depression 55.7% and anxiety 61.8% among patients with chronic medical condition during COVID-19 pandemic [28]. A prevalence of 55.7% was used to achieve the maximum sample size, based on a similar study conducted during the COVID-19 pandemic. By adding a 15% non-response rate, the final sample size was calculated to be 437 using simple proportion formula.

Sampling technique: The average patient load with comorbidities attending treatment at the two selected tertiary hospital OPDs was estimated over the past three months. Based on these patient loads, the sample size was proportionally allocated between the two hospitals. Within each hospital, the sample was further allocated proportionally across the six selected clinics according to their patient volumes. The sampling interval k was calculated by dividing the total average patient load across both hospitals by the overall sample size, yielding a single, common k value. Data were then collected from eligible patients using a systematic random sampling technique by selecting every kth patient across all clinics and hospitals. This approach ensured consistent and proportional representation of patients throughout the study (Fig 1).

Data Collection procedures: A pre-tested, structured interviewer administered questionnaires was used for data collection. The Pretest was conducted by taking 5% of the total sample size prior to the actual data collection period in Yekatit 12 Hospital medical college. Data quality was insured by translating the questionnaire from English to Amharic by considering the culture and norms of the society and then back to English to check the consistency. The data was collected by the primary investigator, two general practitioners, one clinical pharmacist and one nurse. The data were collected under regular supervision after giving training for enumerators about the objective of the study, ethical issues, procedures, techniques and ways of collecting the data. Moreover, COVID-19 preventive measures was taken to ensure protection of the data collectors and the respondents at all-time based on the national and global guidance.

Of the 437 participants approached, 426 completed the survey, yielding a response rate of 97.4%. The remaining 11 individuals did not participate due to refusal. Available demographic data (age, sex, and clinic attended) suggested no systematic differences between responders and non-responders, indicating minimal risk of non-response bias.

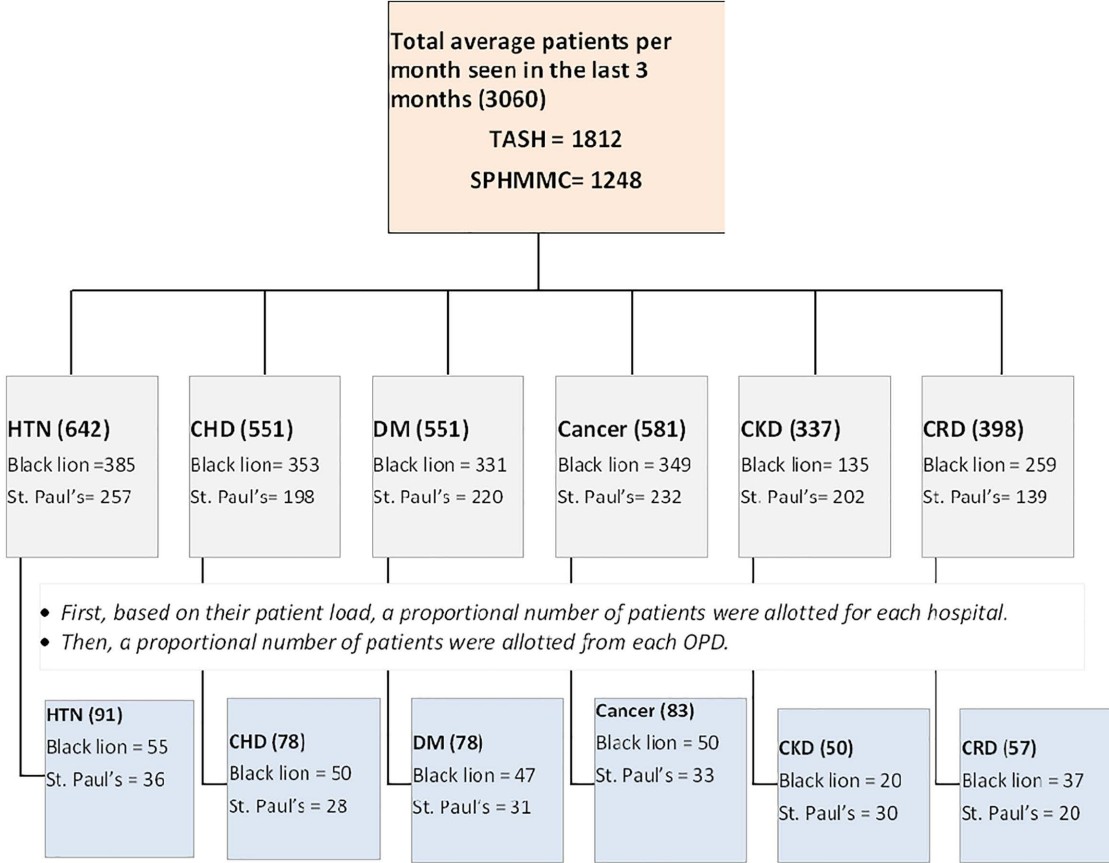

**Fig 1. Schematic diagram for sampling procedure.**

## Data collection tools

The pandemic related perceived stress scale of COVID-19 (PSS-10-C) and Hospital Anxiety and Depression Scale (HADS) are used.

Depression and anxiety were assessed by the Hospital Anxiety and Depression Scale (HADS). This tool has 14 items (seven for each) which, scored in a Likert manner from 0–3, yield a total of 21 points and distinguishes the status of anxiety and depression symptoms. The scale has been validated in different populations in Ethiopia [28–31].

The Oslo-3 social support scale was used to measure social support, with a sum of raw scores ranging from 3 to 114. A score was classified as poor, moderate and strong social support in a range of 3–8, 9–11 and 12–14 respectively [32,33].

The pandemic-related Perceived Stress Scale of COVID-19 (PSS-10-C) was used to assess stress specifically associated with the COVID-19 pandemic. Although the scale has been widely applied internationally, it has not yet been formally validated in Ethiopia, which we acknowledge as a limitation. The PSS-10-C consists of 10 items with five response options ("never," "almost never," "occasionally," "almost always," and "always"). Items 1, 2, 3, 6, 9, and 10 are scored directly from 0 to 4, whereas items 4, 5, 7, and 8 are reverse-scored from 4 to 0 [34,35].

In this study, the PSS-10-C demonstrated satisfactory internal consistency (Cronbach alpha = 0.783). Item–total correlation analysis showed that all items had acceptable values (>0.30), indicating that none of the items performed poorly or required removal.

## Operational definition

Level of depression and anxiety: The 14 item questionnaires using the hospital anxiety and depression scale (HADS) cut-off point was used to determine anxiety and depression level. The (HADS) questionnaire has been well validated. A score of 8/21 is cut-off point for anxiety and depression. The total HADS score is divided into less than 8 (normal), 8–10 (borderline or mild), and 11–21 (abnormal or case) [28–31].

The level of social support was assessed based on 3 item questionnaires using the OSLO social support scale. The OSLO social support scale questionnaire has been well validated. The total score is divided into 3–8 (poor social support), 9–11 (moderate social support), and 1214 (strong social support) [3 2], [33].

The level of COVID-19 related perceived stress: pandemic related perceived stress score cut-off point 27 was used to categorize participants under high perceived stress (≥ 27). Respondents were categorized as follows: low stress: those who scored 0–13 on stress questions; moderate stress: those who scored 14–26 on stress questions; and high perceived stress: those who scored 27–40 on stress questions [34,35].

In this study, multiple comorbidities were defined as the presence of two or more chronic medical conditions in a single patient. The total number of comorbidities per patient was recorded, and descriptive statistics including range and median were calculated. We also explored whether certain combinations of comorbidities were associated with increased psychological distress.

## Data processing, management and analysis

Data consistency and completeness were checked before an attempt was made to enter the code and analyse the data. All questionnaires were checked for completeness at the time of data collection. No missing data occurred for the main outcome variables (PSS-10-C, HADS, Oslo-3). Minimal missing values (<1%) in socio-demographic variables were handled using complete-case analysis.

Data were coded and entered using SPSS (statistical package software for social science) version 25.0 for statistical analysis. STATA 14.1 was also used for data analysis. All of the data were checked for completeness, and internal consistency was also checked to assess the reliability, which gave a Cronbach's alpha = 0.783 for perceived stress scale, and Cronbach's alpha = 0.9 for Hospital anxiety depression scale, which showed good internal consistency for this study. The descriptive analyses of data were presented using numerical summary measures, and the normal distribution of the studied variables was checked, showing that they were normally distributed. Different assumptions were checked prior to performing each statistical analysis. The test of parallel line was checked for ordinal logistic regression. Linear relationship between logarism of the dependent variable and continuous independent variables and multicollinearity were checked for binary logistic regression. The data were presented as frequency tables.

During preliminary data assessment, standard 10-year age intervals were initially considered; however, some of these intervals contained very small numbers of participants. To ensure adequate cell sizes and stable parameter estimates in both ordinal and binary logistic regression models, adjacent age groups were merged. This resulted in the final age categories used in the analysis: 18–38, 39–50, 51–62, and >62.

ordinal logistic regression was used to determine the relationship between perceived stress level and the independent variables; Binary logistic regression analysis with odds ratio along with 95% confidence interval was used to assess the degree of association anxiety and depression had with the independent variables. In bi-variable analysis, variables which have significant association with the dependent variables with p-value of less than 0.25 were entered into multivariate analysis model. This threshold is commonly used in epidemiological studies to reduce the risk of excluding potentially important predictors. A sensitivity analysis using a stricter selection criterion (p < 0.10) was also conducted, and the results remained consistent, confirming the robustness of the variable selection procedure.

Multicollinearity was assessed using Variance Inflation Factor (VIF), tolerance statistics, and Pearson correlation coefficients for continuous variables. Initial diagnostics showed strong multicollinearity among age, duration of illness, and age at

onset; therefore, age at onset was excluded from the final models. All remaining variables demonstrated acceptable multicollinearity levels, with VIF values < 5 and tolerance > 0.2. Model diagnostics were also conducted to assess overall fit. For the binary logistic regression models, the Hosmer–Lemes how goodness-of-fit test ($p > 0.05$), Akaike Information Criterion (AIC), and Nagelkerke pseudo-$R^2$ values were examined. For the ordinal logistic regression model, the proportional odds assumption was confirmed through the test of parallel lines ($p > 0.05$), indicating that the ordinal model was appropriate (S1 Table).

Multivariate logistic regression models using adjusted odds ratios (AORs) were then applied to identify factors associated with psychological outcomes. Statistical significance was declared at $p < 0.05$, and the strength of associations was expressed using AORs with their respective 95% confidence intervals.

## Result

### Socio-demographic characters of participant patients

The data were collected from a total of 426 voluntary patients attending follow up at the outpatient department in selected tertiary Hospitals, Addis Ababa with response rate of 97.4%. The mean age of the participants was 49.57 years (SD ± 15.11 years). Of all participants, 222(52.1%) were females and nearly two-third of participants (68.1%) were currently married. Majority of participants (68.5%) reside in Addis Ababa; 195(45.8%) had more than four families living with them; 107(25.1%) lived in a single room; 90(21.1%) can't read or write, 233(54.7%) were primary care giver for children or other relatives, and 120(28.2%) were housewives by occupation (Table 1).

### Clinical characters and risk assessments of participant patients

Of all study participants, 175 (41.1%) had hypertension; 127 (29.8%) had chronic heart disease (CHD); 125 (29.3%) had diabetes mellitus (DM); 83 (19.5%) had cancer; 81 (19%) had chronic kidney disease (CKD); and 74 (17.4%) had chronic respiratory disease (CRD). Among the participants, 199 (47.3%) had been living with their chronic condition for more than five years, and 205 (48.7%) had an age at illness onset greater than 42 years.

In terms of COVID-19 risk assessment, 50 (11.7%) reported respiratory symptoms, and 65 (15.3%) did not use hand sanitizer. Additionally, 85 (20%) lacked social support in case of isolation or quarantine due to COVID-19. About 290 (68.1%) engaged in physical activity, while 12 (2.8%) reported current substance use. Poor social support was observed in 40.4% of participants (Table 2).

Regarding comorbidity, among the study participants, 198 (46.5%) had multi-morbidity. The number of chronic diseases per patient ranged from 1 to 4, with a median of 1. The most common comorbidity pattern was the coexistence of hypertension and diabetes mellitus (S2 Table).

### Prevalence of perceived stress, anxiety and depression among participants patients with comorbidity

The mean perceived stress score of the respondents was 24.19 (±9.36SD). About 45.5% (95% CI: 40.85-50.31) of the respondents reported severe stress levels, 42.3% (95% CI: 37.63–47.02) reported moderate stress levels, and 12.2% (95% CI: 9.41–15.69) reported low COVID-19 related perceived stress levels.

Accordingly, the prevalence of depression and anxiety during COVID-19 pandemic among patients with comorbidities were 39.7% (95% CI: 35.11-44.42) and 26.5% (95% CI: 22.53-30.94), respectively. These prevalence estimates include both borderline and abnormal cases. The mean depression and anxiety scores were 7.31 ± 5.36 (ranging 0–21) and 5.99 ± 4.91 (ranging 0–21), respectively.

Regarding the level of depression and anxiety, the majority of respondents reported abnormal depression (28.6%) and anxiety (18.1%) level; whereas borderline depression and anxiety was reported in 11% and 8.5% of the patients respectively. These results are detailed in Table 3, which shows both the combined "Yes" category (borderline + abnormal) and the separate levels for transparency. This ensures that the overall reported prevalence (39.7% depression, 26.5% anxiety) corresponds to the sum of borderline + abnormal cases (Table 3).

**Table 1. Socio-demographic characteristic of participant patients at OPD, Addis Ababa, Ethiopia, July 2021.**

| S. No | Variables | Category | Number | Percent |
|---|---|---|---|---|
| 1 | Age of participants | 18-38 | 109 | 25.6 |
| | | 39-50 | 119 | 28 |
| | | 51-62 | 97 | 22.8 |
| | | >62 | 100 | 23.5 |
| 2 | Sex | Male | 204 | 47.9 |
| | | Female | 222 | 52.1 |
| 3 | Marital status | Currently married | 290 | 68.1 |
| | | Currently not married | 136 | 31.9 |
| 4 | Place of residence | Outside Addis Ababa | 134 | 31.5 |
| | | Addis Ababa | 292 | 68.5 |
| 5 | Occupation | House wife | 120 | 28.2 |
| | | Employed | 154 | 36.2 |
| | | Unemployed | 115 | 27 |
| | | Others | 37 | 8.7 |
| 6 | Educational level | Can't read or write | 90 | 21.1 |
| | | Primary | 103 | 24.2 |
| | | Secondary | 109 | 25.6 |
| | | Tertiary | 124 | 29.1 |
| 7 | Family size of the participant patients | 1-4 | 231 | 54.2 |
| | | >4 | 195 | 45.8 |
| 8 | Being primary care giver for family members | Yes | 233 | 54.7 |
| | | No | 193 | 45.3 |
| 9 | Number of rooms in the house | 1 | 107 | 25.1 |
| | | 2 | 119 | 27.9 |
| | | >2 | 200 | 46.9 |

## Associated factors with perceived stress related to COVID -19 among patients with comorbidity

In the bivariate analysis, age, sex, marital status, educational status, occupation, place of residence, number of rooms in the house, CKD, CHD, chronic respiratory illness, diabetics, number of comorbidities, duration of illness, having social support if infected with COVID-19, respiratory symptoms, physical activity and social support were selected for multivariable analysis with p value less than 0.25. Age, marital status, place of residence, diabetics, number of comorbidities, physical activity and social support were selected for the final multivariable ordinal logistic regression model. In the final model, age, place of residence, number of comorbidities, physical activity and social support were significantly associated.

The odds of having high perceived stress compared to moderate perceived stress, low perceived stress is 9.3(95%CI: 4.80, 18.10), 12.7(95%CI: 6.70, 23.95) and 6.1(95%CI: 3.30, 11.30) times greater in those who belong to age groups 18–38, 39–50 and 51–62, respectively compared to those who belong to age group of 63 and above. The odds of having high perceived stress compared to moderate perceived stress, low perceived stress is 1.6(95%CI: 1.03, 2.50) times greater in those who lived in Addis Ababa compared to those who lived outside Addis Ababa. The odds of having high perceived stress compared to moderate perceived stress, low perceived stress is 1.7(95%CI: 1.10, 2.70) times greater in those who had more than one comorbidity compared to those who had only one comorbidity. The odds of having high perceived stress compared to moderate perceived stress, low perceived stress is 1.9(95%CI: 1.20, 2.90) times greater in those who are physically active compared to those who are not. The odds of having high perceived stress compared to moderate perceived stress, low perceived stress is 3.2(95%CI: 1.90, 5.20)

**Table 2. Clinical characteristics and risk assessment of participant patients at OPD, Addis Ababa, Ethiopia, July 2021.**

| S. No | Variables | Category | Number | Percent |
|---|---|---|---|---|
| 1 | Type of chronic disease | CKD | 81 | 19 |
| | | hypertension | 175 | 41.1 |
| | | cancer | 83 | 19.5 |
| | | diabetics | 125 | 29.3 |
| | | CHD | 127 | 29.8 |
| | | CRD | 74 | 17.4 |
| | | Others | 15 | 3.5 |
| 2 | Number of comorbidities | 1 | 228 | 53.5 |
| | | >1 | 198 | 46.5 |
| 3 | Age of onset of illness | <31 | 105 | 24.9 |
| | | 31-41 | 111 | 26.4 |
| | | >42 | 205 | 48.7 |
| 4 | Duration of illness | ≤5 years | 222 | 52.7 |
| | | >5 | 199 | 47.3 |
| 5 | Presence of respiratory symptoms | Yes | 50 | 11.7 |
| | | No | 375 | 88.3 |
| 6 | Contact history | Yes | 44 | 10.3 |
| | | No | 382 | 89.7 |
| | | No | 85 | 20 |
| 7 | Having social support if infected with COVID-19 | Yes | 341 | 80 |
| | | No | 85 | 20 |
| 8 | Use of mask | Yes | 426 | 100 |
| 9 | Use of sanitizer | Yes | 361 | 84.7 |
| | | No | 65 | 15.3 |
| 10 | Physical activity | Yes | 290 | 68.1 |
| | | No | 136 | 31.9 |
| 11 | Substance use | Yes | 12 | 2.8 |
| | | No | 414 | 97.2 |
| 12 | number of people who are so close to you that you can count on them if you have great personal problems | none | 24 | 5.6 |
| | | 1-2 | 191 | 44.8 |
| | | 3-5 | 96 | 22.5 |
| | | 5+ | 115 | 27 |
| 13 | level of much interest and concern people show in what you do | none | 90 | 21.1 |
| | | little | 26 | 6.1 |
| | | uncertain | 19 | 4.5 |
| | | some | 52 | 12.2 |
| | | a lot | 239 | 56.1 |
| 14 | How easy is it to get practical help from neighbors if you should need it | Very difficult | 181 | 42.5 |
| | | Difficult | 33 | 7.7 |
| | | Possible | 39 | 9.2 |
| | | Easy | 52 | 12.2 |
| | | Very easy | 121 | 28.4 |
| 15 | Strength of social support | Poor | 172 | 40.4 |
| | | Moderate | 125 | 29.3 |
| | | Strong | 129 | 30.3 |

**Table 3. Prevalence of psychological problems among participant patients at OPD, Addis Ababa, Ethiopia, July 2021.**

| S. No | Variables | Status | Number | Percent | Level | Number | Percent |
|---|---|---|---|---|---|---|---|
| 1 | Perceived stress level | | | | Low | 52 | 12.2 |
| | | | | | Moderate | 180 | 42.3 |
| | | | | | High | 194 | 45.5 |
| 2 | Anxiety | Yes | 113 | 26.5 | normal | 313 | 73.5 |
| | | | | | Borderline | 36 | 8.5 |
| | | No | 313 | 73.5 | abnormal | 77 | 18.1 |
| 3 | Depression | Yes | 169 | 39.7 | normal | 257 | 60.3 |
| | | | | | Borderline | 47 | 11 |
| | | No | 257 | 60.3 | abnormal | 122 | 28.6 |

**Note:** The "Yes" category for depression and anxiety includes participants with either borderline or abnormal symptoms as defined by the HADS. Percentages for borderline and abnormal levels are reported separately for transparency.

and 1.7(95%CI: 1.02, 2.80) times greater in those who had poor and moderate social support, respectively compared to those who had strong social support (Table 4).

### Associated factors with depression among the patients with comorbidity

In the bivariate analysis, age, sex, marital status, educational status, occupation, being primary care giver for children or other relatives, number of rooms in the house, CKD, hypertension, cancer, diabetics, CHD, number of comorbidities, duration of illness, physical activity and social support were selected for multivariable analysis with p value less than 0.25. Age, marital status, educational status, being primary care giver for children or other relatives, diabetics, CHD, number of comorbidities and social support were selected for the final multivariable logistic regression model using backward LR method. In the final model, age, current marital status, educational status, diabetics, CHD, number of comorbidities and social support were significantly associated with depression.

The odds of having depression among those who belong to age groups 18–38 and 39–50 were 2.8(95% CI: 1.20–50) and 2.6(95% CI: 1.30-5.30) times higher respectively compared to those who belong to age group 63 years and above. The odds of having depression among those who are currently not married were 1.8(95% CI: 1.04–2.95) times higher compared to those who are married. The odds of having depression among those who can't read or write were 2.14 (95% CI: 1.09–4.17) times higher compared to those who completed college and above. Respondents with the lifetime history of diabetes and CHD had 0.5 times the (95% CI: 0.30–0.90) odds of developing depression than those with no history of diabetes and CHD. Respondents with more than one comorbidity had 3.2 times the (95% CI: 1.90–5.40) odds of developing depression than those with only one comorbidity. Respondents with poor social support had 3.7 times the (95% CI: 2.20–6.50) odds of developing depression than those with strong social support (Table 5).

### Associated factors with anxiety among the patients with comorbidity

In the bivariate analysis, age, sex, marital status, occupation, residence, being primary care giver for children or other relatives, number of rooms in the house, CKD, diabetics, chronic respiratory illness, number of comorbidities, duration of illness and social support were selected for multivariable analysis with p value less than 0.25. Age, sex, marital status, being primary care giver for children or other relatives, number of comorbidities and social support were selected for the final multivariable logistic regression model using backward LR method. In the final model, age, sex, being primary care giver for children or other relatives, number of comorbidities and social support were significantly associated with anxiety.

The odds of having anxiety among those who belong to age group 18–38 were 2.6(95% CI: 1.10–5.99) times higher compared to those who belong to age group 63 years and above. The odds of having anxiety among females were

**Table 4. Associated factors with perceived stress level due to COVID-19 among participant patients at OPD, Addis Ababa, Ethiopia, July 2021.**

| variable | Perceived stress | | | COR, 95%CI | AOR, 95%CI | P-value |
|---|---|---|---|---|---|---|
| | Low | Moderate | Severe | | | |
| Age | | | | | | |
| 18-38 | 6 | 40 | 63 | 12.1(6.60,22.40) | 9.3(4.80,18.10) | 0.000 |
| 39-50 | 3 | 39 | 77 | 16.8(9.10,30.95) | 12.7(6.70,23.95) | 0.000 |
| 51-62 | 7 | 45 | 45 | 7.7(4.20,14.20) | 6.1(3.30,11.30) | 0.000 |
| >62 | 35 | 56 | 9 | | 1.00 | |
| Residence | | | | | | |
| Out of Addis Ababa | 24 | 65 | 45 | | 1.00 | |
| Addis Ababa | 28 | 115 | 149 | 2.1(1.40,3.10) | 1.6(1.04,2.50) | 0.033 |
| Current marital status | | | | | | |
| Currently married | 43 | 138 | 109 | | 1.00 | |
| Currently not married | 9 | 42 | 85 | 2.7(1.80,4.10) | 1.4(0.90,2.30) | 0.134 |
| Diabetics | | | | | | |
| No | 36 | 119 | 146 | | 1.00 | |
| Yes | 16 | 61 | 48 | 0.7(0.50,1.10) | 0.8(0.50,1.30) | 0.313 |
| Number of comorbidities | | | | | | |
| 1 | 30 | 102 | 96 | | 1.00 | |
| >1 | 22 | 78 | 98 | 1.3(0.90,1.90) | 1.7(1.10,2.70) | 0.016 |
| Respiratory symptoms | | | | | | |
| No | 50 | 160 | 166 | | 1.00 | |
| Yes | 2 | 20 | 28 | 1.7(0.99,3.10) | 1.5(0.80,2.80) | 0.227 |
| Physical activity | | | | | | |
| No | 25 | 58 | 53 | | 1.00 | |
| Yes | 27 | 122 | 141 | 1.6(1.10,2.40) | 1.9(1.20,2.90) | 0.005 |
| Social support | | | | | | |
| Poor | 10 | 57 | 105 | 3.9(2.50,6.30) | 3.2(1.90,5.20) | 0.000 |
| Moderate | 13 | 62 | 50 | 1.8(1.10,2.90) | 1.7(1.02,2.80) | 0.042 |
| Strong | 29 | 61 | 39 | | 1.00 | |

2.04(95% CI: 1.24–3.37) times higher compared to males. Further, the odds of having anxiety among those who are primary care giver for children or other relatives were 1.88(95% CI: 1.10–3.20) times higher compared to those who are not the primary care giver. Respondents with more than one comorbidity had 1.8 times the (95% CI: 1.11–2.98) odds of developing anxiety than those with only one comorbidity. Lastly, respondents with poor and moderate social support had 3.9(95% CI: 2.04–7.45) and 2.3(95% CI: 1.12–4.64) times the odds of developing anxiety than those with strong social support respectively (Table 6).

## Model diagnostics

The logistic regression models demonstrated acceptable goodness-of-fit. The Hosmer–Lemeshow test was non-significant ($p > 0.05$), indicating that the model fit the data adequately. Model explanatory power was also acceptable, with a Nagelkerke pseudo-$R^2$ of 0.637 in the unadjusted model and 0.629 after adjustment. The Akaike Information Criterion (AIC) was used to compare alternative model specifications, with the final model showing the lowest AIC value.

For the ordinal logistic regression model assessing perceived stress levels (low, moderate, high), the proportional odds assumption was met, as indicated by a non-significant test of parallel lines ($p > 0.05$). This confirmed that the ordinal regression model was appropriate for the analysis.

**Table 5. Associated factors with depression among participant patients at OPD, Addis Ababa, Ethiopia, July 2021.**

| Variable | Depression | | COR, 95%CI | AOR, 95%CI | P-value |
|---|---|---|---|---|---|
| | Yes | No | | | |
| Age | | | | | |
| 18-38 | 53 | 56 | 3.36(1.83-6.14) | 2.78(1.20-5.00) | 0.009 |
| 39-50 | 59 | 60 | 3.49(1.93-6.32) | 2.60(1.27-5.30) | 0.009 |
| 51-62 | 35 | 62 | 2.00(1.07-3.75) | 1.40(0.67-2.80) | 0.409 |
| >62 | 22 | 78 | | 1.00 | |
| Current marital status | | | | | |
| Currently married | 97 | 193 | | 1.00 | |
| Not married | 72 | 64 | 2.24(1.48-3.39) | 1.80(1.04-2.95) | 0.035 |
| Educational status | | | | | |
| Can't read or write | 41 | 49 | 1.52(0.87-2.65) | 2.14(1.09-4.17) | 0.025 |
| Primary | 43 | 60 | 1.30(0.76-2.23) | 1.20(0.70-2.30) | 0.511 |
| Secondary | 41 | 68 | 1.09(0.64-1.87) | 1.00(0.50-1.80) | 0.990 |
| Tertiary | 44 | 80 | | 1.00 | |
| Being primary care giver | | | | | |
| No | 67 | 126 | | 1.00 | |
| Yes | 102 | 131 | 1.46(0.99-2.17) | 1.40(0.80-2.20) | 0.225 |
| Diabetics | | | | | |
| No | 128 | 173 | | 1.00 | |
| Yes | 41 | 84 | 1.52(0.98-2.35) | 0.50(0.30-0.90) | 0.018 |
| Cardiac | | | | | |
| No | 126 | 173 | | 1.00 | |
| Yes | 43 | 84 | 0.11(0.46-1.08) | 0.50(0.30-0.90) | 0.012 |
| Number of comorbidities | | | | | |
| 1 | 76 | 152 | | 1.00 | |
| >1 | 93 | 105 | 1.77(1.19-2.62) | 3.20(1.90-5.40) | 0.000 |
| Social support | | | | | |
| Poor | 101 | 71 | 4.91(2.94-8.19) | 3.70(2.20-6.50) | 0.000 |
| Moderate | 39 | 86 | 1.56(0.89-2.74) | 1.50(0.80-2.70) | 0.227 |
| Strong | 29 | 100 | | 1.00 | |

## Disscusion

In the context of a pandemic, the perceived "threat" can be ubiquitous, potentially carried by anyone in close proximity. From a psychopathological perspective, the COVID-19 pandemic constitutes a relatively novel form of stressor or trauma [36]. Findings from this study clearly demonstrate that the psychological well-being of individuals with comorbidities may be adversely affected by the pandemic and its associated psychosocial consequences. Elevated levels of perceived stress, anxiety, and depression have been consistently linked to the ongoing public health crisis.

The data for this study were collected during the early post–first-wave phase of COVID-19 in Ethiopia, a period marked by high case numbers in April 2021, followed by a gradual decline, with limited vaccine coverage. This context may have intensified psychological distress among chronic patients due to heightened uncertainty, fear of infection, and disruptions in routine healthcare services. Because of these conditions, the levels of stress, anxiety, and depression observed in this study may differ from those measured in later pandemic phases—such as after broader vaccination or improved health-system stability. Nonetheless, the findings offer important insights into mental health vulnerabilities during acute public-health crises and remain relevant for guiding post-pandemic mental health support strategies.

**Table 6. Associated factors with anxiety among participant patients at OPD, Addis Ababa, Ethiopia, July 2021.**

| variable | Anxiety | | COR, 95%CI | AOR, 95%CI | P-value |
|---|---|---|---|---|---|
| | Yes | No | | | |
| **Age** | | | | | |
| 18-38 | 37 | 72 | 4.63(2.15-9.93) | 2.57(1.10-5.99) | 0.029 |
| 39-50 | 39 | 80 | 4.39(2.06-9.36) | 2.26(0.98-5.20) | 0.056 |
| 51-62 | 27 | 70 | 3.47(1.58-7.65) | 1.99(0.86-4.66) | 0.110 |
| >62 | 10 | 90 | | 1.00 | |
| **Sex** | | | | | |
| Male | 36 | 168 | | 1.00 | |
| Female | 77 | 145 | 2.48(1.57-3.90) | 2.04(1.24-3.37) | 0.005 |
| **Current marital status** | | | | | |
| Currently married | 62 | 228 | | 1.00 | |
| Currently not married | 51 | 85 | 2.21(1.41-3.45) | 1.63(0.94-2.82) | 0.08 |
| **Being primary care giver** | | | | | |
| No | 39 | 154 | | 1.00 | |
| Yes | 74 | 159 | 1.84(1.18-2.87) | 1.88(1.09-3.21) | 0.022 |
| **Number of comorbidities** | | | | | |
| 1 | 51 | 177 | | 1.00 | |
| >1 | 62 | 136 | 1.58(1.03-2.44) | 1.82(1.11-2.98) | 0.017 |
| **Social support** | | | | | |
| Poor | 68 | 104 | 4.97(2.68-9.23) | 3.89(2.04-7.45) | 0.000 |
| Moderate | 30 | 95 | 2.40(1.22-4.72) | 2.28(1.12-4.64) | 0.023 |
| Strong | 15 | 114 | | 1.00 | |

A majority of the patients reported moderate to severe levels of perceived stress due to the pandemic, accounting for 42.3% and 45.5%, respectively. This finding aligns with a study conducted in southwest Ethiopia [34], but it is higher than results reported in studies conducted among the general population in Gondar, China, and Turkey [37–40]. The observed variation may be attributed to differences in study populations, as individuals with chronic illnesses tend to experience significantly higher levels of psychological distress. Additionally, the disruption of routine healthcare services for chronic conditions during the COVID-19 outbreak may have served as a persistent source of stress for these patients. Conversely, the reported stress levels in this study were lower than those observed in a similar study conducted in Tigray [34], possibly reflecting disparities in the socioeconomic and political contexts between the two regions. The findings were also lower than those reported in a study from India involving patients undergoing hemodialysis [40], a population known to experience elevated stress levels due to both socioeconomic pressures and ongoing health challenges. Although common stressors associated with chronic illness can be empirically identified, notable differences exist across conditions in terms of the type and severity of stress experienced. Treatment-related demands, such as daily medication adherence, frequent dialysis sessions, and lifestyle restrictions, combined with the inability to fulfill previous social roles, represent significant sources of psychological burden for many patients [41].

It was observed that 39.6% of the patients exhibited depressive symptoms, while 26.6% reported anxiety symptoms during the COVID-19 pandemic. These findings are lower than those reported at Mettu Karl Referral Hospital in Ethiopia, where the prevalence of depression and anxiety among patients with chronic medical conditions was higher [28]. This discrepancy may be attributed to differences in study settings and populations, including variations in literacy levels, socioeconomic status, and access to specialized medical care. The disproportionate economic impact of the pandemic and challenges in accessing healthcare services may have also contributed to the differing outcomes. Additionally, the

lower prevalence observed in the current study may be influenced by social desirability bias. Compared to pre-pandemic data, the prevalence of depression in this study (39.6%) is notably higher than the 5.73% reported prior to COVID-19 [42]. Conversely, the prevalence of anxiety (26.6%) is lower than the 32% reported among patients with chronic medical conditions in Ethiopia before the onset of the pandemic [42]. These variations could be due to differences in study populations, socio-cultural contexts, or the measurement tools used to assess psychological outcomes. Social desirability bias may also partially explain the lower reported rates in the current study. Furthermore, the prevalence of depression and anxiety in the current sample was higher than that reported among the general population in Gondar [37], which is likely due to differences in study populations, as the previous study was conducted among the general public rather than patients with chronic illnesses. In contrast, the current findings are lower than those reported among patients with chronic and immuno-compromising conditions in Saudi Arabia during the COVID-19 pandemic [42], and in other studies involving patients with chronic illnesses [43]. These differences may reflect variations in study settings, cultural and political contexts, and the instruments used to assess depression and anxiety.

This study identified several demographic and clinical factors associated with increased levels of COVID-19-related psychological distress among patients with chronic medical conditions. Notably, individuals aged 62 and above reported significantly lower levels of perceived stress compared to younger age groups. Similar findings have been reported in studies from Iran, where higher stress levels were observed among individuals aged 21–40 [15], likely due to economic uncertainty, job instability, and increased exposure to distressing information via social media among younger populations [44,45]. Additionally, older adults may be more accustomed to staying at home, resulting in less disruption to their daily routines during periods of mandatory self-isolation [34]. In contrast, studies from Ethiopia and Turkey reported higher stress levels among older adults [34,39] possibly reflecting regional differences in healthcare access and the availability of social support systems.

The finding that nearly half of the participants had multiple comorbidities underscores the complexity of managing chronic diseases during the COVID-19 pandemic. Participants residing in Addis Ababa reported significantly higher levels of perceived stress compared to those living outside the city. Moreover, individuals with multiple comorbidities experienced greater stress than those with a single chronic condition, likely reflecting increased health risks, limited access to medications, and heightened vulnerability to severe COVID-19 outcomes [34].

Among the study population, certain comorbidity combinations—particularly hypertension and diabetes mellitus, were associated with the highest stress levels. In this study, multi-morbidity was operationally defined as the presence of two or more chronic conditions, with the number of conditions per patient ranging from 1 to 4 and a median of 1. These findings highlight the importance of identifying high-risk comorbidity profiles to guide the development of targeted mental health interventions for patients with chronic illnesses during the pandemic.

Interestingly, patients who reported engaging in physical activity had higher levels of perceived stress, a finding that contrasts with most existing literature, which generally shows that physical activity reduces stress. This may reflect reverse causation, where more stressed patients intentionally increased their physical activity as a coping strategy. Additionally, the study did not differentiate between types, intensity, or context of physical activity, and some activities (such as outdoor or occupational exercise) might have increased perceived exposure risk to COVID-19, thereby raising stress. This unexpected association warrants further investigation to clarify contextual or measurement influences.

Those with poor or moderate social support also had higher stress levels, potentially driven by fears of isolation or dying alone. These factors can exacerbate anxiety, stress, and depressive symptoms, particularly during public health crises.

Younger people were found to be at higher risk of depression, which aligns with studies from the UK and Iran [44,46]. Depression was also more common among those who were widowed, divorced, or single, possibly due to reduced social support and feelings of loneliness [46]. People with lower levels of education faced a higher risk of depression as well, likely linked to financial struggles and economic stress [46]. Interestingly, a study from Iran showed the opposite:

individuals with higher education reported more depression, which might be due to greater awareness and concern about the situation [44].

Depression and anxiety were more common among patients with multiple comorbidities [37,44,47,48] and those lacking strong social support, in line with findings from Ethiopia, Iran, and China [28,44,49]. These results highlight the added psychological burden experienced by individuals with complex medical conditions and limited emotional support.

Female gender was significantly associated with higher levels of anxiety during the pandemic, consistent with findings from studies conducted in Iran, the UK, and Saudi Arabia [42,44,46].

Younger age also emerged as a strong predictor of anxiety, as supported by multiple international studies [42,44,46]. Additionally, being a primary caregiver was linked to increased anxiety, likely due to heightened concern for the well-being of dependents. Similarly, a study from Iran found that individuals living with children experienced higher levels of anxiety compared to those living with other adults [46].

Overall, the findings of this study align with existing literature, underscoring the psychological vulnerability of younger individuals, women, and patients with multiple comorbidities or limited social support during the COVID-19 pandemic. Targeted psychosocial interventions are crucial for these high-risk groups to reduce the risk of long-term mental health consequences.

Handling multicollinearity was important to ensure valid estimates in the regression analyses. Age, age at onset of illness, and duration of illness demonstrated very high correlations, and therefore age at onset was removed from the final models to improve model stability. This adjustment strengthened the reliability of the regression results and ensured that individual predictors contributed meaningfully to the overall model.

The acceptable results of the goodness-of-fit tests further support the validity of the models. The proportional odds assumption for the ordinal regression model was met, indicating that perceived stress levels followed a consistent pattern across predictor categories. Together, these diagnostic procedures confirmed that the models used were appropriate and yielded interpretable findings regarding psychological outcomes among chronic patients during the COVID-19 pandemic.

## Limitations of the study

This cross-sectional study cannot establish temporal relationships, and reliance on self-reported data may have introduced inaccuracies and social desirability bias. Data collection relied on interviewer-administered questionnaires, as many chronic-disease patients were older adults with limited literacy, making self-administered questionnaires impractical. While this ensured comprehension and inclusion, it may have increased the risk of social desirability bias, potentially leading to under reporting of psychological symptoms. Interviewers were trained to maintain neutrality, ensure confidentiality, and encourage honest responses, and the questionnaire included validation items to improve reliability. The use of only quantitative methods without qualitative triangulation limited deeper understanding of participants' perspectives. Additionally, patients with severe mental health issues who may have stopped attending hospitals could have been missed, possibly underestimating the prevalence of mental health problems. Research design and researcher bias might have influenced outcomes. Given the fluctuating nature of pandemic-related mental health impacts, longitudinal studies are needed to better capture these changes over time.

We acknowledge that multiple statistical comparisons were performed in this study without formal correction for multiple testing. This may increase the likelihood of Type I errors; therefore, the findings, particularly those approaching significance, should be interpreted with caution. Further research with larger samples and confirmatory analyses is recommended to validate these results.

This study has notable limitations related to selection bias. First, excluding critically ill patients and those experiencing acute exacerbations may have led to underestimation of psychological distress, as individuals in acute crisis often experience heightened anxiety and stress. However, we made this exclusion intentionally because acute medical episodes, such as DKA, HHS, or severe asthma/COPD exacerbations, are themselves powerful psychological stressors that would make it difficult to isolate the mental health impact attributable specifically to the COVID-19 pandemic. Second,

the exclusion of patients with pre-existing mental health conditions likely also contributed to underestimation of the true prevalence of distress, as these individuals may have been more vulnerable during the pandemic. In addition, our sample included only patients who continued to attend outpatient clinics, meaning that individuals who avoided care due to fear, financial constraints, or severe psychological distress were not captured.

We lacked data on participants' or family members' COVID-19 infection status, income loss, employment disruption, family deaths, healthcare disruptions (e.g., missed appointments), detailed socioeconomic indicators (e.g., household income), access-to-care variables (e.g., travel distance, granular residence), and chronic disease duration/severity—all of which may influence psychological outcomes. Substance use prevalence (2.8%) is likely underestimated due to social desirability bias, and 100% reported mask use probably reflects reporting bias. The term "primary caregiver" is defined in Methods as main responsibility for day-to-day care of a dependent household member. Finally, protective factors (resilience, coping strategies, religious/spiritual support) were not assessed. Future studies should address these gaps by collecting comprehensive pandemic-related, socioeconomic, clinical, and protective factor data.

Finally, because the study was conducted in two tertiary referral hospitals in the capital city, generalizability to rural populations or lower-level health facilities is limited.

## Conclusion

The COVID-19 pandemic has exacerbated psychological distress among patients with chronic comorbidities in Addis Ababa, with high prevalence of perceived stress, depression, and anxiety observed. Younger age, multiple comorbidities, poor social support, and female gender were key factors associated with increased mental health burden. These findings highlight the urgent need for targeted psychosocial interventions and support services tailored to vulnerable groups. Given the study's cross-sectional nature, further longitudinal research is essential to better understand the evolving mental health impact of the pandemic on this high-risk population.

## Supporting information

**S1 Table. Assessment of multicollinearity among independent variables.**
(XLSX)

**S2 Table. Distribution of comorbidities among study participants.**
(XLSX)

**S1 Data. Dataset.**
(XLSX)

## Acknowledgments

I sincerely thank my advisors for their invaluable guidance. I am grateful to the staff and administration at Yekatit 12 Hospital, Tikur Anbessa Specialized Hospital, and Saint Paul's Hospital Millennium Medical College, as well as the patients who participated in this study.

Special thanks to Ms. Kalkidan Dege, Ms. Yordanos Dege, Mr. Ananiya Tilahun, Dr. Maraki Mehari, Dr. Amanuel Asres, Dr. Tewodros Haile, and Dr. Muluken Tesfaye for their valuable support and advice.

Lastly, I deeply appreciate my family and friends for their unwavering encouragement.

## Author contributions

**Conceptualization:** Etsegenet Dege Dires, Ayto Addisu Negash.

**Data curation:** Etsegenet Dege Dires, Ayto Addisu Negash, Getachew W/Yohannes, Muna Ahmed Redi, Birhanu Fenta Tsega, Agazhe Melaku Mihiretie, Sifessa Dessalegn Demshasha, Tekiy Markos Bedore, Yitayew Ewnetu Mohammed.

**Formal analysis:** Etsegenet Dege Dires, Ayto Addisu Negash, Getachew W/Yohannes, Firdos Ali Mohammed, Birhanu Fenta Tsega, Agazhe Melaku Mihiretie, Sifessa Dessalegn Demshasha, Tekiy Markos Bedore.

**Funding acquisition:** Etsegenet Dege Dires, Getachew W/Yohannes, Sifessa Dessalegn Demshasha.

**Investigation:** Etsegenet Dege Dires, Ayto Addisu Negash, Firdos Ali Mohammed, Muna Ahmed Redi, Birhanu Fenta Tsega, Sifessa Dessalegn Demshasha, Tekiy Markos Bedore, Yitayew Ewnetu Mohammed.

**Methodology:** Etsegenet Dege Dires, Ayto Addisu Negash, Getachew W/Yohannes, Muna Ahmed Redi, Agazhe Melaku Mihiretie, Yitayew Ewnetu Mohammed.

**Project administration:** Etsegenet Dege Dires, Ayto Addisu Negash, Getachew W/Yohannes, Birhanu Fenta Tsega, Sifessa Dessalegn Demshasha.

**Resources:** Etsegenet Dege Dires, Ayto Addisu Negash, Firdos Ali Mohammed, Muna Ahmed Redi, Agazhe Melaku Mihiretie, Sifessa Dessalegn Demshasha, Tekiy Markos Bedore.

**Software:** Etsegenet Dege Dires, Ayto Addisu Negash, Birhanu Fenta Tsega.

**Supervision:** Etsegenet Dege Dires, Ayto Addisu Negash, Getachew W/Yohannes, Firdos Ali Mohammed, Muna Ahmed Redi, Yitayew Ewnetu Mohammed.

**Validation:** Etsegenet Dege Dires, Ayto Addisu Negash, Getachew W/Yohannes, Muna Ahmed Redi, Agazhe Melaku Mihiretie, Tekiy Markos Bedore, Yitayew Ewnetu Mohammed.

**Visualization:** Etsegenet Dege Dires, Ayto Addisu Negash, Firdos Ali Mohammed, Birhanu Fenta Tsega.

**Writing – original draft:** Etsegenet Dege Dires, Ayto Addisu Negash, Getachew W/Yohannes, Birhanu Fenta Tsega, Agazhe Melaku Mihiretie, Tekiy Markos Bedore.

**Writing – review & editing:** Etsegenet Dege Dires, Ayto Addisu Negash, Getachew W/Yohannes, Firdos Ali Mohammed, Muna Ahmed Redi, Birhanu Fenta Tsega, Agazhe Melaku Mihiretie, Sifessa Dessalegn Demshasha, Tekiy Markos Bedore, Yitayew Ewnetu Mohammed.

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
