## [Decision Letter · Decision Letter 0]

8 Dec 2025

PGPH-D-25-03264

Prevalence and Determinants of Depression, Anxiety, and Perceived Stress Among Patients with Chronic Comorbidities Attending Outpatient Clinics in Addis Ababa During the COVID-19 Pandemic

Dear Dr. Negash,

Thank you for submitting your manuscript to PLOS Global Public Health. After careful consideration, we feel that it has merit but does not fully meet PLOS Global Public Health’s publication criteria as it currently stands. Therefore, we invite you to submit a revised version of the manuscript that addresses the points raised during the review process.

We look forward to receiving your revised manuscript.

Kind regards,

Theingi Maung Maung, Ph.D

Academic Editor

Journal Requirements:

1. We have amended your Competing Interest statement to comply with journal style. We kindly ask that you double check the statement and let us know if anything is incorrect.

2. Please provide separate figure files in .tif or .eps format.

3. Your manuscript is missing the following sections: Discussion. Please ensure these are present, and in the correct order, and that any references to subheadings in your main text are correct. An outline of the required sections can be consulted in our submission guidelines here:

https://journals.plos.org/globalpublichealth/s/submission-guidelines#loc-parts-of-a-submission

4. We have noticed that you have uploaded Supporting Information files, but you have not included a list of legends. Please add a full list of legends for your Supporting Information files after the references list.

Reviewers' comments:

Reviewer's Responses to Questions

**Comments to the Author**

1. Does this manuscript meet PLOS Global Public Health’s publication criteria?

Reviewer #1: Yes

Reviewer #2: Yes

2. Has the statistical analysis been performed appropriately and rigorously?

Reviewer #1: No

Reviewer #2: Yes

3. Have the authors made all data underlying the findings in their manuscript fully available (please refer to the Data Availability Statement at the start of the manuscript PDF file)?

Reviewer #1: Yes

Reviewer #2: Yes

4. Is the manuscript presented in an intelligible fashion and written in standard English?

Reviewer #1: Yes

Reviewer #2: Yes

Reviewer #1: The paper deals with a significant issue of the mental health effects of COVID-19 on chronic conditions in Ethiopia. The research is overall well-constructed and offers useful information on a population that is not much studied. Nevertheless, there are a number of methodological and presentation concerns that should be resolved before publication.

1. The data were collected between April- June 2021, which is almost four years ago. Although the authors do not dismiss this as a cross-sectional study, there is little information on the impact of the pandemic phase of data collection (post-initial wave, pre-vaccination in Ethiopia) on the generalizability. The discussion must be clear:

• . What is the stage of the pandemic in this?

• The way the findings could vary in other phases of the pandemic.

• The applicability of these results to mental health assistance following the pandemic.

2. The criterion of the exclusion of critically ill patients or patients with acute exacerbation can be a great source of selection bias. Psychological distress may be the greatest in patients who have acute exacerbations. The authors should:

• . Measure the number of patients who were not included for this reason.

• Talk about how this omission may result in the underestimation of prevalence.

• This is one of the significant weaknesses in generalizability.

3. Social desirability bias is cited as one of the possible limitations by the authors, but it is not fully dealt with. Considering the nature of face-to-face interviews and the culture:

• What did interviewers do to reduce this bias?

• Did it contain validation questions?

• Did the authors ever think of using anonymous self-administered questionnaires?

4. The paper gives 46.5% as having had multiple comorbidities, but Table 3 presents a lot of patients with their conditions overlapping. Please clarify:

• What was the method of counting multiple comorbidities (e.g., a patient with hypertension, diabetes, and heart disease = 3 conditions)?

• What was the range, median number of comorbidities?

• Was it that some of the combinations were more closely linked with psychological distress?

5.

a) Multivariate Model Variable Selection: The authors selected p<0.25 in bivariate analysis to select the variables. Although this method is common, it might have irrelevant variables. Could you explain why this threshold was chosen, and this should be analyzed by sensitivity analysis with p<0.10.

b) Multicollinearity: The authors state that they have verified the presence of multicollinearity, although they do not provide the values of VIF. Since there is a likelihood of correlation between age, duration of illness, and age at onset, please provide:

• Final models of VIFs of all variables.

• Continuous variable correlation matrix.

c) Model Fit Statistics No goodness-of-fit statistics are provided on the logistic regression models (e.g., Hosmer-Lemeshow test, AIC, pseudo-R 2). These should be included.

d) Ordinal Logistic Regression: Did proportional odds assume low/moderate/high levels of perceived stress? The authors state that they do the test of parallel lines, but they do not give the results.

6. The fact that stress was reported higher among physically active patients (AOR=1.9, p=0.005) is counterintuitive and should be given some more consideration. The concise explanation of the authors (an increased risk of exposure) is speculative. Consider:

• Is this reverse causation (stressed patients working out to get better)?

• Were there variations in kinds of physical activity?

• This observation is contrary to the literature, and it requires more in-depth investigation.

7. Figure 1 indicates k=7, though it is referred to as systematic random sampling. Please explain whether k was different by clinic or not.

• The rate of respondents was very good (97.4), and what about the 11 non-responders? Were they systematically dissimilar?

8. The PSS-10-C is said to be pandemic-related, but we should verify [34, 35]. Was this particularly confirmed in Ethiopia regarding COVID-19?

9. The Cronbach alpha of PSS-10-C is not high but satisfactory (0.783). Bad item-total correlations?

a) Table 1 & 3 (this should be Table 2): The numbering of the table seems wrong. The second table is actually Table 3 on page 15.

b) Age Categories: The age categories (18-38, 39-50, 51-62, >62) do not seem to be arbitrary. What was the rationale? It is better to use common age groups (e.g., 10-year intervals) to make the study more comparable to others.

c) Missing Data: No discussion of missing data. Are there any declined responses? How were they handled?

10. The abstract indicates severe stress (45.5%), depression (39.7%), and anxiety (26.5%).

11. Table 5 (Table 12 in the list), however, displays the depression and anxiety as yes/no, but also by level.

12. In their reporting of depression at 39.7, is this combined with borderline (11%) + abnormal (28.6)? Please clarify throughout.

13.

a) Comparison with Pre-pandemic Data: You compare with a pre-pandemic Ethiopian study [42], but it would be nice to see more pre-pandemic Ethiopian baseline data in the discussion, provided that it exists.

b) Clinical Implications: The prevalence and determinants have received much discussion, although little guidance is given on:

• Which particular interventions can be performed in the conditions of resource limitation?

• What is the way of incorporating mental health screening into chronic disease clinics?

• What are the implications of the health system?

c) Contradictory Results: In the age and stress discussion, you have support of younger adults experiencing higher stress (Iran) and older adults experiencing higher stress (Ethiopia, Turkey). The discussion will be enhanced by increasingly questioning why you have found a correlation with certain studies rather than others.

14. The limitations are properly recognized; however, the limitations are to be added with:

• Absence of pre-pandemic baseline data for the same patients.

• Failure to determine the status of COVID-19 infection or loss of family and loved ones.

• None of the past mental health history (not within treatment).

• One-point assessment in a dynamic pandemic.

• Page 4: "2740% of patients with chronic obstructive lung disease (COPD)- initial mention, COPD needs to be spelled out completely, followed by the abbreviation.

• Page 5: the lifestyle modifications should be lifestyle modifications.

• Page 6: please spell out (Gregorian Calendar?) GC on June 30, 2021GC

• Unstable abbreviations: OPD vs outpatient department.

15. You identified scores as poor (3-8), moderate (9-11), and strong (12-14). Nonetheless, only 3 items will be rated on a scale of 1-5, so it should be 15 instead of 14. Please confirm that your scoring system is equivalent to your references [32, 33]. The discussed methods are the ways to evaluate environmental, socio-economic, and health-related issues related to COVID-19, although these data are not provided in the results. Were these collected? In that case, they would be a useful context.

16. You have applied 55.7% prevalence in [28] and at the same time 22.8% in [27]. Why choose 55.7%? A smaller sample would be required to make a more conservative estimate. This move has to be explained.

Essential Revisions:

• The numbering of tables is correct.

• Explain the count of multiple comorbidities and give descriptive statistics.

• All regression model fit statistics Report models.

• Give VIF values to show that there is no multicollinearity.

• Be more expansive on the context of the time and the generalizability.

• Discuss the physical activity paradox in more detail.

• Include clinical, health system implications in the discussion.

• Be more specific on how selection bias can be a problem.

• Add information on COVID-19-related stressors in case they were gathered.

• Take into consideration stratified analysis based on the type of chronic condition.

Reviewer #2: General Comments

This study addresses an important topic regarding the mental health impact of COVID-19 on patients with chronic comorbidities in Ethiopia. The research is timely and contributes to understanding vulnerable populations during the pandemic. However, there are several limitations and areas requiring clarification or improvement.

Major Limitations and Concerns

1. Temporal Validity and Relevance

Data collection period (April-June 2021) represents a specific pandemic phase that may not reflect current or evolving mental health impacts

The cross-sectional design captures only a snapshot during one pandemic wave, limiting understanding of how psychological distress evolved over time

Consider acknowledging that findings may be specific to that pandemic period and discussing how mental health trajectories may have changed since

2. Selection Bias and Generalizability

Exclusion of patients with pre-existing mental health conditions creates significant selection bias. These patients likely experienced the most severe psychological impact during COVID-19, yet they were systematically excluded

This exclusion means your prevalence estimates are likely underestimates of the true burden

The sample includes only patients who continued attending outpatient clinics. Patients who stopped seeking care due to fear, financial constraints, or severe psychological distress were missed

Both hospitals are tertiary referral centers in the capital city, limiting generalizability to rural populations or secondary healthcare settings

3. Social Desirability Bias

You acknowledge this limitation but do not discuss its potential magnitude

Face-to-face interviews may have led participants to underreport psychological symptoms, particularly in a culture where mental health stigma exists

Self-administered questionnaires might have reduced this bias

4. Lack of Comparison Group

No pre-pandemic baseline data for the same population

No concurrent control group (e.g., patients without chronic conditions or general population)

This makes it difficult to attribute findings specifically to the pandemic versus chronic illness burden alone

5. Causality and Temporal Relationships

The cross-sectional design precludes establishing whether chronic conditions preceded or followed mental health symptoms

Cannot determine if COVID-19 stress caused the observed mental health outcomes or if pre-existing distress was simply measured during the pandemic

Methodological Concerns Requiring Clarification

6. Sampling and Response Rate

While you report a 97.4% response rate (426/437), this seems exceptionally high for voluntary participation in mental health research

Please clarify: Were patients approached consecutively? How many refused before the interview began?

The systematic sampling assumes all eligible patients attended during the study period—how did you handle missed appointments or irregular attendance patterns?

7. Multiple Comorbidities and Disease Complexity

46.5% had multiple comorbidities, but the analysis doesn't account for specific comorbidity combinations

Different chronic disease combinations may have differential psychological impacts (e.g., cancer + CKD vs. hypertension + diabetes)

Disease severity and functional status were not assessed

8. Measurement Issues

PSS-10-C (COVID-specific perceived stress):

You report using PSS-10-C but provide limited validation information for the Ethiopian context

Was this tool translated and validated specifically for Amharic speakers?

The cut-off of ≥27 for "high stress" needs better justification—is this validated in your population?

HADS:

While validated in Ethiopian populations, most citations refer to HIV patients or adolescents—different from your chronic disease population

Cultural adaptation details are missing

Oslo-3:

Very brief measure (3 items) of a complex construct

May not capture pandemic-specific changes in social support networks

9. Statistical Analysis Concerns

Multicollinearity: You mention checking for it but don't report VIF values or tolerance statistics

Model fit statistics: No goodness-of-fit indices reported for logistic regression models

Multiple testing: With numerous comparisons, no correction for multiple testing is mentioned (though this may be acceptable in exploratory research)

Effect sizes: Only odds ratios presented—consider reporting standardized effect sizes for better interpretability

10. Missing Pandemic-Specific Variables

No assessment of COVID-19 infection history among participants or family members

Vaccination status not captured (though this may predate widespread vaccination in Ethiopia)

No measure of pandemic-related losses (income, employment, family deaths)

Healthcare disruption not quantified (missed appointments, delayed treatments)

Minor Limitations and Suggestions

11. Incomplete Characterization of Sample

Median household income or poverty indicators not reported

Distance traveled to hospital not assessed (relevant for access and stress)

Urban vs. rural origin within "outside Addis Ababa" not differentiated

Specific chronic disease duration and severity not detailed

12. Physical Activity Paradox

Your finding that physically active patients had higher stress (AOR=1.9) is counterintuitive and inadequately explained

Was this possibly reverse causation (anxious patients exercising more to cope)?

Physical activity assessment appears to be a simple yes/no—intensity, duration, and type not captured

13. Data Quality Concerns

2.8% substance use seems very low for a population under stress—possible underreporting

100% mask use is suspiciously high—suggests social desirability bias

Some variables (e.g., "being primary caregiver") are not clearly defined

14. Limited Exploration of Protective Factors

Study focuses primarily on risk factors

Resilience, coping strategies, and positive adaptation not assessed

Religious/spiritual support not evaluated despite mentioning religious festival restrictions as stressors

15. Language and Presentation

Some repetition between results and discussion sections

Tables could be streamlined (Table 2 is very long)

Statistical notation inconsistencies (CI presentation varies)

Recommendations for Authors

For Discussion Section:

More explicitly address the selection bias from excluding patients with pre-existing mental health conditions and those who stopped attending clinics

Discuss the physical activity paradox more thoroughly—this is a surprising finding that needs explanation

Acknowledge that prevalence estimates are likely conservative due to multiple sources of underestimation

Expand discussion of cultural context—how might Ethiopian cultural norms around mental health expression have influenced findings?

Provide more context on healthcare system disruptions in Ethiopia during this period

For Limitations Section:

The current limitations section is brief. Consider expanding to include:

Selection bias from excluding patients with mental health conditions

Inability to distinguish pandemic-specific effects from chronic illness burden

Lack of pre-pandemic baseline or control group

Potential social desirability bias in face-to-face interviews

Limited generalizability to rural populations and lower-level health facilities

Snapshot nature of cross-sectional data during one pandemic phase

Unmeasured confounders (disease severity, functional status, economic impact)

For Future Research Recommendations:

Longitudinal studies tracking mental health trajectories

Inclusion of patients with pre-existing mental health conditions

Mixed-methods research incorporating qualitative data

Intervention studies testing psychosocial support programs

Comparison with pre-pandemic cohorts

Assessment of long-term mental health consequences

Specific Questions for Authors

Why were patients with pre-existing mental health conditions excluded? This seems to exclude the most vulnerable group.

Can you provide validation data for PSS-10-C in the Ethiopian context?

How did you handle patients with multiple comorbidities in the analysis? Were interaction effects explored?

What was the refusal rate before consent? The 97.4% response rate seems unusually high.

Can you explain the counterintuitive finding regarding physical activity and increased stress?

Were any interventions offered to patients identified with severe depression, anxiety, or stress?

**Do you want your identity to be public for this peer review?** For information about this choice, including consent withdrawal, please see our Privacy Policy

Reviewer #1: **Yes: ** Dr. Jonah Bawa Adokwe

Reviewer #2: **Yes: ** Tooba Adil

---

## [Decision Letter · Decision Letter 1]

30 Dec 2025

Prevalence and Determinants of Depression, Anxiety, and Perceived Stress Among Patients with Chronic Comorbidities Attending Outpatient Clinics in Addis Ababa During the COVID-19 Pandemic

PGPH-D-25-03264R1

Dear Dr Negash,

We are pleased to inform you that your manuscript 'Prevalence and Determinants of Depression, Anxiety, and Perceived Stress Among Patients with Chronic Comorbidities Attending Outpatient Clinics in Addis Ababa During the COVID-19 Pandemic' has been provisionally accepted for publication in PLOS Global Public Health.

Best regards,

Theingi Maung Maung, Ph.D

Academic Editor

Reviewer Comments (if any, and for reference):

Reviewer's Responses to Questions

**Comments to the Author**

Reviewer #1: All comments have been addressed

publication criteria?

Reviewer #1: Yes

3. Has the statistical analysis been performed appropriately and rigorously?

Reviewer #1: Yes

4. Have the authors made all data underlying the findings in their manuscript fully available (please refer to the Data Availability Statement at the start of the manuscript PDF file)?

Reviewer #1: Yes

5. Is the manuscript presented in an intelligible fashion and written in standard English?

Reviewer #1: Yes

Reviewer #1: I recommend acceptance for publication.

**Do you want your identity to be public for this peer review?** For information about this choice, including consent withdrawal, please see our Privacy Policy

Reviewer #1: **Yes: ** Dr. Jonah Bawa Adokwe
